# Leukemia Mortality among Benzene-Exposed Workers in Brazil (2006–2011)

**DOI:** 10.3390/ijerph20136314

**Published:** 2023-07-07

**Authors:** Maria Juliana Moura-Corrêa

**Affiliations:** Department of Environmental Health Surveillance and Occupational Health, Health and Environmental Surveillance Secretariat, Ministry of Health (DSAST/SVSA/MS), SRTVN 701, Via W5 Norte, Brasilia 70719-040, Brazil; maria.juliana@saude.gov.br; Tel.: +55-61-3315-3653

**Keywords:** occupational exposure 1, benzene 2, leukemia 3, mortality 4, carcer

## Abstract

Background: In this study, the annual leukemia mortality rate is estimated by occupational groups potentially exposed to benzene in Brazil and compared to non-exposed workers by sex. Methods: Data were extracted from the Mortality Information System and the National Institute of Geography and Statistics from 2006 to 2011. Occupational groups exposed to benzene were defined by using the Finnish Job-Exposure Matrix, FINJEM. Results: We found 21,049 leukemia deaths in 1917 in occupational groups potentially exposed to benzene, corresponding to an annual average mortality rate of 4.5/100,000, higher than the estimate for non-exposed workers: 2.6/100,000, corresponding to a Mortality rate ratio MRR = 1.7. Each benzene-exposed occupational group had increased leukemia mortality, and printers and occupations in graphics presented the highest MRR (2.7), followed by laboratory assistants (MRR = 2.6), laundry workers, chemists, and upholsterers, each of these occupational groups presenting MRR = 2.3. Conclusions: Benzene shows the need for better enforcement of protective norms against this known carcinogen. Our results support the need for better enforcement of protective norms to reduce benzene exposure.

## 1. Introduction

Neoplasia is a major cause of death in the world. In 2013, from a total of 8.2 million neoplasia-related deaths, approximately 265,000 were leukemia cases. Leukemia mortality was estimated at 4.8/100,000 in countries with stable economies, higher than the 3.5/100,000 estimate reported for developing countries [1]. One major occupational-related cause of leukemia is benzene exposure. Findings from a retrospective follow-up study carried out in China showed that, among benzene-exposed workers, the leukemia mortality rate was 14/100,000 person-years, much higher than in the unexposed group (2/100,000 person-years). Although the estimation of mortality rates for specific trades was limited by the small numbers, results suggest that workers from the painting industry had the highest leukemia mortality rate (15.9/100,000 person-years) [2]. This study was expanded, and the new data analysis showed a borderline mortality rate ratio (MRR) for the association of all types of neoplasia and benzene-exposed workers (MRR = 1.2; 95% Confidence Interval, CI: 1.0–1.5) and for leukemia limited to males (MRR = 2.1; 95% CI: 1.0–5.3). For both sexes, benzene exposure was associated with deaths from all combined hematolymphoproliferative cancers (MRR = 2.6; 95% CI: 1.5–5.0) and leukemia (MRR = 2.6; 95% CI: 1.3–5.7), particularly the acute myelogenous type (MRR = 3.1; 95% CI: 1.2–10.7) [3]. Subsequently, data from this study for the period 1972 to 1999 were analyzed and showed MMR = 2.8; 95% CI: 1.6–5.5 [4]. In contrast, researchers from the Dow Chemical Company [5] analyzed industry data combined with death certificates and did not find evidence that benzene exposure correlates with leukemia deaths. This finding was interpreted as a result of the availability of and access to better health care leading to increased life expectancy and declining mortality rates or the healthy worker effect.

Benzene is not the only risk factor for leukemia, which opens the question of how many leukemia deaths can be attributable to benzene exposure. In the US, using an estimated relative risk of 2.0 and 4.0, the attributable fractions of leukemia deaths to benzene (AFLB) were 0.8% and 2.0%, respectively [6], both higher than Great Britain’s estimate of 0.14% [7]. There are sex differences in the AFLB, with men having a higher AFBL (0.7%) compared to women (0.2%) [8], which is a common pattern: male-prevailing occupations present higher levels of exposure.

In Brazil, the age-adjusted leukemia mortality rate was estimated at 2.7/100,000 in 1980 and 2.5/100,000 in 1995 for the general population [9]. No occupation or trade-specific estimates were found. A recent study estimated the number of workers potentially exposed to benzene by occupational groups based on the Finnish National Job-Exposure Matrix (FINJEM) after matching their codes to those of the Brazilian Classification of Occupations [10]. In the present study, we used these data and death records to estimate the average annual leukemia mortality rate among benzene-exposed and non-exposed groups by each potentially exposed occupational group. In addition, we estimated the corresponding number of leukemia cases attributable to this exposure.

## 2. Materials and Methods

This is a mortality study carried out with data from Brazil’s Mortality Information System (SIM). Leukemia comprises all deaths coded C90.0 through C95.9 in the International Classification of Diseases, 10th Revision (ICD-10). The distribution of workers by occupational group is provided by the Brazilian Institute of Geography and Statistics (IBGE), regardless of the type of job contract, i.e., formally registered or having informal jobs. Based on this occupational distribution and on FINJEM [11], the number of workers potentially exposed to benzene at the workplace was estimated [10] and then utilized to calculate leukemia mortality rates for each occupational group. Because of changes in the occupational coding system, the study time was limited to the period 2006 to 2011. Furthermore, because latency time may vary from 0 to 20 years and the study focuses on work-related exposure to benzene, the population was limited to individuals aged 18 years or older.

Leukemia cases correspond to the underlying cause registered in death certificates of all leukemia types and myelodysplastic syndromes, analyzed through the six ICD-10 groups: C90, multiple myeloma, and malignant plasma cell neoplasms; C91, lymphoid leukemia; C92, myeloid leukemia; C93, monocytic leukemia; C94, other leukemias of specified cell type; and C95, leukemia of unspecified cell type. In Brazil, occupations have been registered in death certificates through codes of the Brazilian Classification of Occupations-2002 (CBO-2002), based on the International Standard Classification of Occupations (ISCO-8812), since 2006 [12]. In the present study, the FINJEM occupational groups were utilized, which enabled the use of the estimated number of benzene-exposed workers in Brazil [10]: (1) Chemists; (2) Laboratory assistants; (3) Service station attendants; (4) Upholsterers; (5) Leather cutters for footwear; (6) Shoe sewers; (7) Lasters and sole fitters, not elsewhere classified (nec); (8) Footwear workers, nec; (9) Machine and engine mechanics; (10) Painters, lacquerers and floor layers; (11) Printers; (12) Occupations in graphics, (13) Distillers; (14) Cookers, furnacemen (chemical processes); (15) Refinery/occupations in chemical industry; (16) Rubber product workers; (17) Laundry workers. According to FINJEM, workers from these occupational groups are potentially exposed to benzene because the Finish available workplace measurements were equal to or higher than 0.1/ppm on average over one year [11]. Cases from the remaining occupational groups were considered free of work-related benzene exposure. The descriptive variables were sex and age groups analyzed in quintiles: 18–39, 40–53, 54–64, and 65 years or over.

### Analysis

Leukemia cases were described according to types and socio-demographic variables. The average leukemia yearly mortality rate was estimated by dividing the total number of leukemia deaths during the study time by the sum of the estimated number of workers for each calendar year. To prevent underestimation of mortality rates, leukemia cases with missing occupations were distributed proportionally across the occupational groups of the cases that had valid occupation registers. The population attributable fraction (PAF) to occupational benzene exposure of leukemia deaths was estimated using the Levin equation [13]: PAF = P × (MRR − 1)/[P × (MRR − 1) + 1], where P is the prevalence of occupational exposure to benzene in Brazil, and MRR is the ratio of leukemia mortality (exposed to non-exposed). As all cases were analyzed, statistical tests were not used. The analysis was carried out using the software SAS 9.2. The study protocol was registered in the Brazilian National Research Ethics System and approved by the Internal Review Committee of the Institute of Collective Health, Federal University of Bahia (Protocol No. 17268313.8.0000.5030).

## 3. Results

There were 21,049 leukemia deaths registered during the study time. Valid occupations were registered in only 7888 records (37.5%), of which 5449 were male (69.1%), and 2439 were female cases (30.9%). Of the 7888 records, 718 (9.1%) were from the potentially occupational benzene exposure group, and 7170 (90.9%) were from the non-exposed group. Table 1 shows that occupationally benzene-exposed cases were more likely to be male, younger, and have a myeloid leukemia diagnosis as the underlying cause of death compared to the non-exposed group. The proportion of cases increases as calendar time increases, regardless of benzene exposure. For both males and females, the distribution of leukemia deaths by type did not show major differences across occupational benzene exposure groups, except for myeloid leukemia, which is more common in the exposed group (Table 2). 

Based on all leukemia cases (*n* = 21,049), the average annual crude mortality rate (LMR—leukemia mortality rate) was 2.7/100,000 workers, calculated based on the total number of cases, providing that those with missing occupations were proportionally allocated to each occupational group. In the potentially exposed group, LMR was 4.5/100,000, higher than the 2.6/100,000 estimate calculated for the non-exposed group, with a crude leukemia mortality rate ratio (LMRR) of 1.7 (Figure 1). The highest LMR (6.9/100,000 workers) was estimated for printers and occupations in graphics, followed by laboratory assistants (6.7/100,000), laundry workers (6.1/100,000), chemists and upholsters (6.0/100,000), and painters, lacquerers, and floor layers (5.1/100,000), among others. LMRR estimates show that LMR is higher in each exposed occupational group compared to the non-exposed group (Figure 1). Overall, LMR for males (2.9/100,000) was slightly higher than LMR for females (2.5/100,000), LMRR = 1.2 (Table 3). In each occupational group, men have higher LMR than women, except for laundry workers. In addition, among male workers, the risk of leukemia death was 5.0/100,000, higher for upholsterers (8.5/100,000), printers and occupations in graphics (8.2/100,000), and chemists (7.9/100,000). For women, the overall LMR estimate was 3.2/100,000. However, unlike men, the highest LMR among female workers was obtained for laundry workers (6.7/100,000), laboratory assistants (6.5/100,000), painters, lacquerers, and floor layers (4.1/100,000). All LMRRs for women workers were smaller than the corresponding ones for men (Table 3).

The population attributable fraction (PAF) of leukemia deaths to occupational benzene exposure was 6.2%, which corresponded to 118 attributable cases out of 1917 expected cases among all workers. Males had higher PAF and attributable deaths (6.2%, *n* = 108) compared to women (PAF = 0.5%, *n* = 10). PAF estimates were small, except for machine and engine mechanics: 69 leukemia cases (PAF = 6.6%) were likely to be caused by occupational benzene exposure (Figure 2). 

## 4. Discussion

Our findings showed that the workers who died from leukemia due to occupational exposure to benzene were more likely to be male and young and that the underlying cause of death was myeloid leukemia. Workers potentially exposed to benzene had a higher LMR compared to the non-exposed, independently of their occupational group and sex. This confirms evidence of benzene’s carcinogenicity, particularly for leukemia. The impact of occupational benzene exposure was greater among men, who had a higher PAF compared to women; consequently, a larger number of preventable leukemia cases was found among men. 

The predominance of young males aged 39 years or younger in leukemia death cases was reported by Demian et al. [14], who showed that working males are relatively young compared to the general population. This is suggestive of low latency time, given that the legally established age for labor onset is 18 years. Therefore, for a 20-year maximum latency, exposed workers in the age group 18–39 years have an increased risk of benzene-related occupational leukemia death [15]. Males prevail in industrial trades and industry-related occupations, and it is known that they endure long-standing chemical exposures, such as that of benzene, among other work-related risk factors [16]. However, the higher occurrence of the disease in men does not allow us to draw conclusions regarding the influence of sex on risk. One possible hypothesis is that women may have jobs in which benzene exposure is lower [17]. In Brazil, norms for workers’ protection and actions against benzene exposure at the workplace are poorly enforced [18]. It should be noted that apprentices can enter the labor market at the age of 15, but only in non-dangerous occupations; however, this might not be adopted in remote or poor areas. Several initiatives were carried out, such as surveillance by the institutions responsible for the protection of health and work in the occupational and environmental spheres, regulations, the valorization of the participation of workers in risk prevention actions, intervention in territory with exposure, campaigns, information, and risk communication on the carcinogenicity of benzene [19]. 

The most common type of underlying cause of death recorded was myeloid leukemia, which comprises approximately 70.0% of all cases among benzene-exposed workers. Our results also showed that in each occupational group with potential benzene exposure, LMR was higher compared to the non-exposed category, regardless of leukemia type. This is consistent with the conclusions of the IARC Monograph 100 F [20], which states that there is sufficient evidence of benzene’s carcinogenicity for acute myeloid leukemia and acute non-lymphocytic leukemia. In addition, several studies have found positive associations between benzene and acute and chronic lymphocytic leukemia, non-Hodgkin lymphoma, and multiple myeloma. Benzene metabolites generate multiple genotoxic effects in hematopoietic stem cells, causing chromosomal abnormalities reported among cases of cancer in hematopoietic tissues. In addition, leucopenia and chromosomal damage—early markers of hematopoietic cancer—have been found in workers exposed to benzene. Our findings were clearer for myeloid leukemia, presumably because of the larger number of cases. 

The high number of deaths from acute myeloid leukemia corroborates the findings of other studies [21,22]. Regarding all leukemias, we showed that the group with the highest LMR was the printers and graphics one, which was different from Linet’s findings [4]. However, our classification of occupational groups was different from that of the Linet study. Analyzing all chemical groups together, including chemical manufacturing, petroleum refining, and marketing and distribution of benzene-exposed chemicals, the number of leukemia deaths in this large group would be higher than in the other occupational groups, according to the investigations carried out in China [4] and the United States [23]. As well as bringing together all the chemical groups of chemical manufacturing, petroleum refining, marketing, and distribution of benzene-exposed chemicals, leukemia will be higher in these workers than in others, according to observations from the investigations in China [4] and the United States [23]. Therefore, when all the chemical occupational groups are analyzed together, mortality due to leukemia is even higher.

In our study, males had a higher LMR compared to women in all the analyzed occupational groups except for laundry workers. Occupational benzene exposures are common in petroleum [24,25], rubber [26,27], shoes [17], laundry [28], printing, and gas retail trades [29], among others where males prevail. Similarly, high levels of benzene exposure have been reported for occupational groups related to industrial operation and maintenance [18]. The benzene risk trends reported in these studies agree with our findings.

The PAF estimated for males and females was higher in our study compared to data from England, Finland, and the United States of America, when the relative risk was assumed to be 2.0 [6]. The number of female leukemia deaths was small or non-existent in some occupational groups, which made it impossible to estimate the attributable fraction of each occupational group.

Other factors that can affect the estimation of PAF in workers are the proportion of exposed workers and the uncertainty regarding the exposure magnitude. To correct this error, adjustments concerning workers’ turnover and work shifts [30,31] have been made, especially for exposure to carcinogens that have a long latency period. Some occupations of groups of workers potentially exposed to benzene are usually identified by the characteristics of intense turnover, like service station attendants and work shifts, especially in the chemical and petrochemical industry. However, it was not possible to adjust for turnover in the present study, as we could not find national turnover estimates by occupation groups, only by economic trades [32]. 

The study period was limited to 2006–2011 because, in 2006, CBO-2002 became the occupation classification standard. Additionally, other operational changes were implemented in death certificates and in the SIM due to the large underreporting of occupation in death certificates, which reached 40% in 2013 [33]. 

Conclusions based on the data of our study must be viewed with caution due to uncertainties and limitations, which derive mainly from missing occupation and from possible misdiagnosis or underreporting of deaths due to leukemia. This is particularly worrying when we take three factors into account: the complexity of the leukemia diagnosis, the difficulties in accessing health services, and the quality of the health information record. Occupation, the variable employed to recognize groups of workers potentially exposed to benzene, was not registered in a high number of records. Despite the adjustments that were made, which presumably reduced this problem, the number of deaths may be higher in each one of the groups of interest.

The main limitation of this study refers to possible errors regarding inconsistency in the workers’ data due to differences between the Classification of Occupation in Domicile, COD, CBO, and Demographic Census classifications compared to FINJEM, which prevented us from drawing a direct correspondence between the occupations’ codes. However, to overcome this problem, specialists were asked to evaluate the nominal correspondence between the COD and FINJEM classifications and the CBO 2002 classification. Errors of inconsistency in the workers’ data based on COD were identified in some occupational groups, and, in these cases, an adjustment was made using data from the occupational groups corresponding to the Demographic Census branches of economic activities. In addition, we found inconsistencies in the identification of the subgroup Retail Trade Shoemakers, which composes the category of Leather Cutters for Footwear, whose codification was the same for 58 retail trade occupations. In this situation, the variable was treated as missing data.

Another limitation of this approach was the lack of available data about the prevalence of benzene exposure. Consequently, it is likely that the utilization of the parameters of the FINJEM database to estimate prevalence contributed to underestimating the results. Occupational exposure to benzene can be higher than the one estimated for these workers based on FINJEM, as the exposure may encompass other occupational groups that are not listed in FINJEM’s original group of exposed workers, like taxi drivers, firefighters, and traffic guards, among others. Furthermore, other occupations whose exposures were below 0.1 ppm and were not selected for the exposed groups in Finland may have higher exposures in Brazil. 

The results of the present study refer to severe cases of leukemia that led to death. During the study period, there was a worldwide reduction in mortality from leukemia deriving from significant improvements in treatment, which contributed to an increase in the survival of individuals [1]. Therefore, the number of deaths from severe leukemia cases reported here is significant and can be even higher if we consider the large number of death certificates in which the occupation variable was not registered and the frequent codification problems in conformity with CBO in the SIM. 

As for the magnitude of the exposure and the occurrence of leukemia among the exposed workers, a restriction occurred because only mortality data were used. In addition, leukemia deaths belong to a heterogeneous group with distinct survival periods due to variability according to type and subtype, stage of the diagnosis, prognosis, evolution, and age [34]. Another issue is the use of PAF, which has been criticized by researchers who discuss its methodological limitations. Boffeta [35], for example, argues that, for environmental carcinogenesis, the estimation of this measure depends on the magnitude of cancer due to a certain exposure and the estimated frequency of this exposure. Therefore, errors can occur due to the polarization or confounding effect on the direction of the association force and on the frequency or level of exposure, and these, in turn, can affect the corresponding estimate of the attributable risk. Another limitation is related to the difficulty in calculating the effect of each factor separately, which does not allow one to determine its impact on PAF. However, despite the uncertainty regarding the interval between factors, this can hardly justify differences in the order of magnitude. 

Despite these limitations, this study presents advances in the production of original national estimates of mortality from leukemia related to occupational exposure to benzene in Brazil. Thus, it contributes to the world’s effort to produce estimates of the burden of neoplasia attributable to occupational factors 7,14. In Brazil, the magnitude of occupational exposure related to mortality from leukemia is still insufficiently recognized as an important factor in general mortality. Its prioritization may influence the conduction of population-based studies and raise awareness concerning the importance of registering occupations in the appropriate field in information systems. Consequently, it can improve the quality of the data collected in national studies, which is essential to the production of reliable estimates.

In the present study, the utilization of the occupation title as a variable indicating exposure was an important methodological definition to estimate potential exposures to benzene and cases of attributable leukemia in workers. Its advantages are related mainly to the identification of risk groups and to the possibility of drawing comparisons with other studies, as the occupation title is widely used in epidemiological studies, and its international standardization enables the establishment of correspondence. In addition, its application reduces the classification bias and provides information to subsidize preventive actions and monitor exposures by occupational group, as such exposures can be perfectly related to branches of economic activities. 

From this perspective, the results of this study show evidence of excess mortality from leukemia among workers potentially exposed to benzene when compared to the other workers in the population. Therefore, our results support the surveillance of occupational neoplasia, considering occupational determinants, risk groups, and measures to prevent and monitor leukemia caused by occupational exposure to benzene. 

As these types of leukemia are, to a large extent, preventable, the study results represent, especially to public health managers, an important source of information that enables them to know the burden attributed to benzene exposure, to guide planning actions to reduce risk factors and to monitor prevention strategies targeted at workers’ health. Especially concerning strategies to reduce the occurrence of deaths from leukemia caused by benzene exposure, the PAF enables us to estimate how many cases would not have occurred if the exposure had been eliminated. 

The potential for application of such analyses has guided countries in the establishment of legal norms to control benzene exposure, either environmental or occupational, and the surveillance of potentially exposed workers.

## 5. Conclusions

This study showed evidence of excess mortality from leukemia among workers, that chronic exposure to benzene. Benzene shows the need for better enforcement of protective norms against this known carcinogen. Our results support the need for better enforcement of protective norms to reduce benzene exposure.

## Figures and Tables

**Figure 1 ijerph-20-06314-f001:**
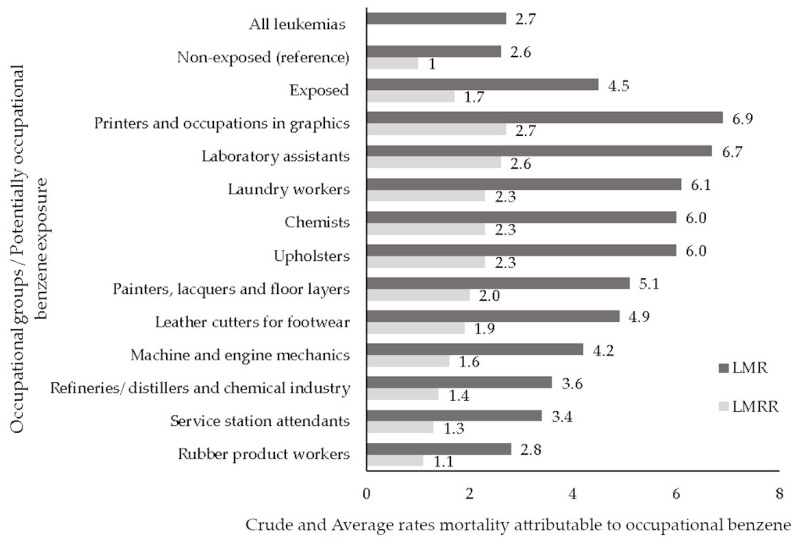
Average annual crude leukemia mortality rate (LMR) and ratios (LMRR) by occupation and potentially occupational benzene exposure. Brazil, 2006–2011.

**Figure 2 ijerph-20-06314-f002:**
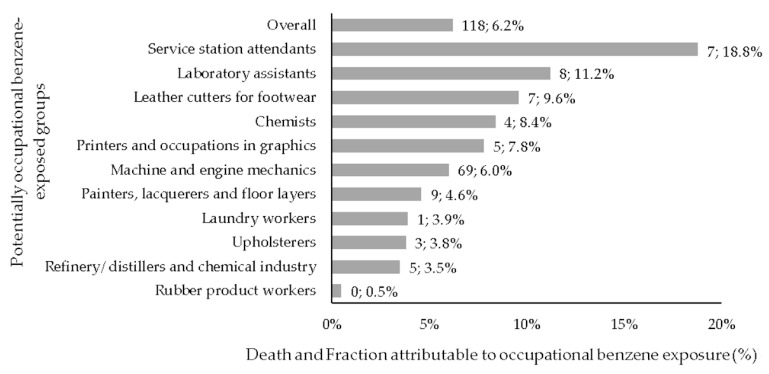
Leukemia attributable fraction to potentially occupational benzene exposure (%) and corresponding number of attributable deaths. Brazil, 2006–2011.

**Table 1 ijerph-20-06314-t001:** Characteristics of leukemia deaths with valid occupation registers (*n* = 7888) by potentially occupational benzene exposure. Brazil, 2006–2011.

Variables	Potentially Occupational Benzene Exposure				Total	
	Exposed		Non-Exposed			
	*N* = 718	100.0%	*N* = 7170	100.0%	*N* = 7888	100.0%
Sex						
Men	564	78.6	4885	68.1	5449	69.0
Women	154	21.4	2285	31.9	2439	31.0
Age groups						
18–39	176	24.5	1.406	19.6	1.582	20.1
40–53	145	20.2	1.482	20.7	1.627	20.6
54–64	149	20.8	1.444	20.1	1.593	20.2
65 or older	248	34.5	2.838	39.6	3.086	39.1
Calendar year						
2006	75	10.4	926	12.9	1001	12.7
2007	109	15.2	1046	14.6	1155	14.6
2008	117	16.3	1235	17.2	1352	17.1
2009	136	18.9	1185	16.5	1321	16.8
2010	146	20.3	1339	18.7	1485	18.8
2011	135	18.8	1439	20.1	1574	20.0
Leukemias and myelodysplastic syndrome types						
C90 Multiple myeloma/plasma cell						
malignant neoplasms	5	0.7	38	0.5	43	0.5
C91 Lymphoid leukemia	130	18.1	1341	18.7	1471	18.7
C92 Myeloid leukemia	512	71.3	4997	60.7	5509	69.8
C93 Monocytic leukemia	7	1.0	55	0.8	62	0.8
C94 Other leukemias/specified cell type	12	1.7	80	1.1	92	1.2
C95 Leukemia of unspecified cell type	52	7.2	659	9.2	711	9.0

**Table 2 ijerph-20-06314-t002:** Leukemia deaths by potentially occupational benzene exposure according to sex. Brazil, 2006–2011.

Sex/Leukemia Types	Potentially Occupational Benzene Exposure				Total	
	Exposed		Non-Exposed			
	*N* = 718	100.0%	*N* = 7170	100.0%	*N* = 7888	100.0%
Men						
leukemias and myelodysplastic syndrome types	564	100.0	4885	100.0	5449	100.0
C90 Multiple myeloma/plasma cell malignant neoplasms	4	0.7	25	0.5	29	0.5
C91 Lymphoid leukemia	104	18.4	998	20.4	1102	20.2
C92 Myeloid leukemia	395	70.0	3313	67.8	3708	68.1
C93 Monocytic leukemia	5	0.9	37	0.8	42	0.8
C94 Other leukemias/specified cell type	12	2.3	54	1.1	66	1.2
C95 Leukemia of unspecified cell type	44	7.8	458	9.4	502	9.2
Women						
leukemias and myelodysplastic syndrome types	154	100.0	2285	100.0	2439	100.0
C90 Multiple myeloma/plasma cell malignant neoplasms	1	0.6	13	0.6	14	0.6
C91 Lymphoid leukemia	26	16.9	343	15.0	369	15.1
C92 Myeloid leukemia	117	76.0	1684	73.7	1801	73.8
C93 Monocytic leukemia	2	1.3	18	0.8	20	0.8
C94 Other leukemias/specified cell type	0	--	26	1.1	26	1.1
C95 Leukemia of unspecified cell type	8	5.2	201	8.8	209	8.6

**Table 3 ijerph-20-06314-t003:** Average annual crude leukemia mortality rate (LMR) and ratios (LMRR) by occupation, sex, and potentially occupational benzene exposure. Brazil, 2006–2011.

Occupational Groups/Potentially Occupational Benzene Exposure	Men		Women	
	Average Annual Crude Mortality/100,000	Crude Mortality Rate Ratio	Average Annual Crude Mortality/100,000	Crude Mortality Rate Ratio
Overall	2.9		2.5	
Potentially occupational benzene exposure				
Non-exposed	2.7 (ref.)	1.0	2.5 (ref.)	1.0
Exposed	5.0	1.9	3.2	1.3
Chemists	7.9	2.9	1.3	0.5
Laboratory assistants	6.7	2.5	6.5	2.6
Service station attendants	3.7	1.4	1.9	0.8
Upholsterers	8.5	3.1	3.7	1.5
Leather cutters for footwear	6.0	2.2	3.0	1.2
Machine and engine mechanics	4.8	1.8	3.0	1.2
Painters, lacquerers, and floor layers	5.1	1.9	4.1	1.6
Printers and occupations in graphics	8.2	3.0	1,7	0.7
Refinery/distillers and chemical industry	3.8	1.4	3.1	1.2
Rubber product workers	3.0	1.1	---	---
Laundry workers	5.6	2.1	6.7	0.8

## Data Availability

Data are contained within the article. Additionally, the data which support the findings of our study are publicly available datasets from the Instituto Brasileiro de Geografia e Estatística (IBGE) and Sistema de Informação sobre Mortalidade (SIM).

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
