# Peer review of "Leukemia Mortality among Benzene-Exposed Workers in Brazil (2006–2011)"

_ijerph, 2023, doi:10.3390/ijerph20136314_

Round 1
Reviewer 1 Report
Methods can be improved and presentation of results can include the statistics or % of deaths in each group; The authors considered the limitations of the study and clearly presented them. The article has value presening the occupation based exposure and deaths due to leukemia.
Author Response
Please see the attachment for response of the reviewer’s comments point-by-point.
We would like to thank the reviewer for evaluating our manuscript. All changer are in red and we have tried to address all concerns properly and we believe that our article has improved considerably.

Reviewer 2 Report
I found the manuscript titled "Leukemia Mortality Among Benzene-exposed Workers in Brazil" by Maria Juliana Moura-Correa to be both interesting and well-written. However, I would like to raise the following points for consideration:
a) As the population investigated was enrolled between 2006 and 2011, the Author should better explain whether precautions have been assumed during the last years.
b) It would be valuable to compare the study's findings with those from other countries to determine any overlaps.
c) Grammatical and spelling errors have been found throughout the manuscript that may warrant attention.
Author Response

(The authors gave the same response as above.)

Reviewer 3 Report
The article of Moura-Correa examined risk of mortality from leukemia in working persons exposed to benzene compared to persons not exposed. Data from a national registry from 2006-2011 were analyzed. Analyzed data showed that the workers who died from leukemia due to occupational exposure to benzene were more likely to be male and young, and that the underlying cause of death was acute myeloid leukemia.
Overall the paper is well written, comprehensive and adds important confirmatory data to prior studies reinforcing the need for protective measures in benzene exposed workers.
I have some minor remarks:
1. Multiple myeloma is not leukemia and should be removed from analysis or multiple myeloma should be specifically mentioned as “leukemias and multiple myeloma”.
2. Discussion is very extensive and should be shortened.
Author Response

(The authors gave the same response as above.)

Reviewer 4 Report
In the manuscript “Leukemia Mortality Among Benzene-exposed Workers in Brazil,” the author described data extracted from the mortality System and IBGE system to show leukemia deaths that could be associated with occupational benzene exposure from the period of 2006 to 2011.
The author showed that the cases of leukemia, especially acute myeloid leukemia, were higher in those exposed to benzene through their work than in others not exposed to benzene. Also, the author showed which professionals have more risks based on the leukemia cases registered in the database.
The major concern about the study is the period that the data is about: from 2006 to 2011. More than ten years have passed since 2011, and the scenario regarding benzene exposure could be different today. If the author could have expanded the research and data from more recent years, we could have more reliable data and information about the current status of occupational exposure to benzene continues to cause the same amount of leukemia cases or has improved since then.
Therefore, the manuscript would offer no new insights into the field of leukemia today.
Author Response

(The authors gave the same response as above.)

Reviewer 5 Report
The study estimate if occupational exposure to benzene increases the leukemia mortality rate of workers in Brazil. Specific comments are listed below:
(1) Benzene is a known agent for carcinogenesis. I guess the authors want to make the novel point of looking for factors that contribute to leukemia mortality rate in certain populations. If the study wants to estimate if occupational exposure to benzene, age, gender etc. are potential factors for the leukemia mortality rate in potentially exposed workers, some statistic tests need to be done to show the comparison and significance.
(2) Why did the data for the study only cover 2006-2011? That is more than 10 years from now. In other words, it may not very well represent the most recent situation.
Author Response

(The authors gave the same response as above.)

Reviewer 6 Report
The manuscript reported leukemia mortality study in benzene-exposed worker in Brazil, with a focus on presenting the fact that the potential correlation between leukemia mortality and benzene-expose. Overall, the manuscript is very simple and lacks novelty. Below is the major concerns:
1. The dataset in this paper is form 2006 to 2011. Given the fact that it is submitted in 2023, the data being used is very old and it is better to update the dataset (at least to 2018).
2. Figures will be more straightforward then tables. It is encouraged that the author can make some plots to highlight the observations.
3. It is interesting that myeloid leukemia is the highest among all other leukemia. Author needs to address the potential reason and also could analyze if there is any difference of mortality rate across different types of leukemia.
4. Is there any dose dependence of mortality rate and possible uptaken amount from different occupations?
5. The statistic approach was not clearly explained. Author should do some statistic test and see the significance rather then saying "higher" or "lower".
Author Response

(The authors gave the same response as above.)

Round 2
Reviewer 2 Report
The Authors have improved the manuscript. No supplementary information are requested.
Reviewer 4 Report
Dear authors,
Thank you for your modifications in the present study. I understand the limitations of the study and agree with the changes made in the manuscript.
Reviewer 5 Report
The authors' efforts to address the reviewer's comments are acknowledged.
Reviewer 6 Report
Authors have addressed all my concerns. I don't have further questions.